# Impact of Phosphogypsum on Viability of *Trichuris suis* Eggs in Anaerobic Digestion of Swine Manure

**DOI:** 10.3390/microorganisms13051165

**Published:** 2025-05-21

**Authors:** Olexandra Boyko, Viktor Brygadyrenko, Yelizaveta Chernysh, Viktoriia Chubur, Hynek Roubík

**Affiliations:** 1Department of Parasitology and Veterinary and Sanitary Expertise, Dnipro State Agrarian and Economic University, Serhii Efremov St., 25, 49000 Dnipro, Ukraine; boikoalexandra1982@gmail.com; 2Department of Biodiversity and Ecology, Oles Honchar Dnipro National University, Nauky Av., 72, 49010 Dnipro, Ukraine; brigad@ua.fm; 3Department of Sustainable Technologies, Faculty of Tropical AgriSciences, Czech University of Life Sciences Prague, Kamýcká 129, 16500 Prague, Czech Republic; chuburv@ftz.czu.cz; 4Department of Ecology and Environmental Protection Technologies, Sumy State University, Kharkivska St., 116, 40007 Sumy, Ukraine

**Keywords:** phosphogypsum, anaerobic digestion, sulfate reduction, digestion, *Trichuris suis*, Nematoda, disinfection

## Abstract

Waste from livestock farms contains various pathogens, including eggs and larvae of helminths—pathogens of parasitic diseases harmful to animals and humans. One of the methods for their effective processing to obtain biofertilizer and biofuel is anaerobic digestion, which requires further improvement to completely suppress the viability of pathogenic microorganisms in mesophilic conditions. To this end, the use of anaerobic digestion under sulfate reduction conditions to suppress pathogens using biogenic hydrogen sulfide is promising. Consequently, this study aims to study the effect of a sulfur-containing additive such as phosphogypsum on the disinfection of pig manure during anaerobic digestion. Egg mortality was already found to increase significantly compared to the control (80% and more), even at a minimum concentration of phosphogypsum (5%), on the fifth day of the experiment. At the same time, the maximum effect (100% mortality of *Trichuris suis* eggs) was recorded at a 10% concentration of phosphogypsum, starting from the 10th day of the study. Our experiment showed that changes in anaerobic digestion conditions using phosphogypsum could positively affect digestate disinfection. However, further research is needed to optimize the conditions of the process for an effective combination of disinfection with the production of environmentally safe organic fertilizers and high-quality biogas with a high level of methane.

## 1. Introduction

Every year, a large amount of organic waste is generated worldwide [1]. One of the most common of these in agriculture is pig waste [2]. Every year, an increase in the demand for pork from extensive livestock farming is recorded. Therefore, preventing pig diseases, including helminthiasis, is of great economic importance. One of the most common parasitic diseases in pigs is trichuriasis. It causes severe diarrhea, anorexia, and loss of productivity due to a decrease in average daily gain in animals [3]. Pig infection occurs when helminth eggs are ingested with feed or water, where they enter the body and develop to the invasive stage within several weeks. The most favorable conditions for their development are pastures, where timely removal of animal excrement is impossible [4]. Therefore, parasite eggs develop actively and can persist in environmental objects for a long time. The development of eggs of the genus *Trichuris* (Nematoda, Trichuridae) in the environment is influenced by many environmental factors, including temperature [5,6]. Scientists have found that with increasing temperature, the time to form invasive eggs (the formation of larvae in the egg) decreases. An experiment was conducted on eggs of *Trichuris skrjabini* (Baskakov, 1924)—parasites of sheep. The most favorable temperature for the development of nematode eggs of this species in laboratory conditions was 25 °C. Thus, on the 54th day, 80.3% of eggs already had mobile larvae.

In general, several publications have focused on the mesophilic anaerobic digestion regime and the efficiency of the methanogenic association in this regime. For example, the study by Rusanowska et al. [7] focuses on organic loading rates rather than specific operating parameters at 36 °C. As stated in Kabaivanova et al. [8] operational parameters such as temperature, pH, and hydraulic retention time significantly affect microbial communities in anaerobic digestion at 36 °C. These communities are critical for optimizing biogas production and process efficiency through their interactions and responses to environmental stressors. Lee et al. [9] focus on a 37 °C anaerobic reactor and emphasize that microbial communities, particularly Clostridium butyricum, Candidatus Roizmanbacteria and Spirochaetaceae, significantly affect methane production and process efficiency, especially at different proportions of food waste in the feedstock.

Consequently, the mesophilic regime (in particular, 36–37 °C) is widely used for the process of the anaerobic digestion of organic waste. However, it is also important to study the effect of this regime on the process of pathogen suppression, which was also indicated in our previous work [10].

Similar studies on the development of eggs in the environment depending on temperature were also conducted in *Trichuris vulpis* (Froelich, 1789). Yevstafieva and Dolhin [11] found that the development period of *Trichuris* eggs to the invasive stage at a temperature of 23–29 °C was 15–27 days. At the same time, at a temperature of 23 °C, 74.0% of eggs had mobile larvae within 27 days. At a temperature of 25 °C, the process of egg development to the invasive stage was shortened and was 24 days (77.7% of the eggs were viable). Therefore, with increasing temperature, the period of development of these pathogens at this stage of the eggs gradually decreased. It was 18 days at a temperature of 27 °C, and 15 days at 29 °C. At the same time, for eggs of *Trichuris suis* Schrank, at a temperature of 27 °C, 1788 were found to reach invasive stages in 40 days and go through seven stages of embryogenesis. Yevstafieva et al. [12] found significant differences in the growth and development of eggs isolated from the gonads of female nematodes and feces of sick animals. The eggs of *Trichuris suis*, which are released by animals into the environment, are more adapted to environmental conditions. Their viability was greater than 96.6% compared to eggs isolated from the gonads of female nematodes. Therefore, such organic waste can pose a threat both to the environment where parasites develop and persist, and to animals that regularly make contact with environmental objects [13,14].

Today, there are many methods for the decontamination of organic waste, including those aimed at destroying helminth eggs and larvae [15,16]. One of the most common and environmentally acceptable methods is anaerobic digestion [17,18]. This method not only disinfects waste, but also produces valuable biofuel [19]. However, the disinfection efficiency during anaerobic disinfection does not always reach 95%, especially when the process is carried out in the mesophilic mode (35–40 °C), which is more economically advantageous [20,21,22]. Therefore, the issue of improving this method to increase the level of disinfection of organic waste [23,24], in particular, the effect of hydrogen sulfide on parasitic disease pathogens and pathogenic microorganisms, which can be produced by sulfate-reducing bacteria under anaerobic conditions, is relevant, taking into account its distant impact on microbiofilm formation and the regulation of inflammatory processes [25,26,27].

A separate area of research is the use of phosphogypsum in biotechnological processes. Phosphogypsum is a grayish powdery mass of varying humidity. It is a compound that includes SO_3_, CaO, SiO_2_, P_2_O_5_, Al_2_O_3_, Fe_2_O_3_, MgO, and fluorides (36%, 26%, 15%, 1.2%, 0.3%, 0.2%, 0.01%, and 0.1–0.4%, respectively) and is a waste generated during the production of phosphate fertilizers [28]. The problem with phosphogypsum processing and storage is that it is associated with environmental pollution, making it a relevant topic of research for many countries of the world [29,30]. Chernysh et al. [30] analyzed publications on the use of phosphogypsum in various industries and identified several clusters. One of the clusters identified by analyzing literary sources was the use of phosphogypsum in agriculture as an ameliorant and fertilizer component, as well as the impact of phosphogypsum on microorganisms, in particular, in bioremediation processes. Onopriienko et al. [31] substantiated the possibility of simultaneously solving the environmental problem of using a large amount of phosphogypsum waste and agrotechnical processes to eliminate the salinization of irrigated soils.

Recent research has focused on the effects of phosphogypsum on the sulfate-reduction process, highlighting its potential as a substrate for sulfate-reducing bacteria (SRB) and various methods for optimizing sulfate reduction. Therefore, the study by Alla et al. [32] explores the decomposition of phosphogypsum, primarily composed of calcium sulfate dihydrate, using metallic iron in hydrochloric acid. It details how sulfate ions from phosphogypsum are reduced to sulfide through various reaction mechanisms. Bounaga et al. [33] discussed optimizing phosphogypsum sulfate leaching to improve biological reduction by SRB. The study reports that five different SRB consortia showed significant reduction activity when using phosphogypsum leachate as a sulfate source. In particular, the highest reduction rate was achieved with lactate as the electron donor, highlighting the biological potential of PG in the generation of hydrogen sulfide. The results of a study by Azabou et al. [34] evaluating the biodegradation capabilities of mixed SRB cultures in phosphogypsum indicate that these bacteria can effectively reduce sulfate concentrations, contributing to the sustainable management of this industrial waste. Alsanea et al. [35] investigated how sulfate leached from phosphogypsum is transformed in biological systems. This research emphasizes the importance of batch operations in increasing the generation of soluble sulfide through the reduction of biological sulfate rather than its precipitation as calcium sulfate.

Further optimization studies [33,36] have been conducted to improve the efficiency of phosphogypsum sulfate leaching, which is crucial to improve bioreduction processes in bioreactors. These studies focus on the composition of the microbial community and the effectiveness of different anaerobic microbial inocula to promote the reduction of sulfate.

However, most of these studies have focused on the following:-The reduction of phosphogypsum to calcium sulfide;-The effect of alkaline leaching on sulfate reduction;-Sulfate reduction using mixed cultures;-Sulfate leaching and transformation.

Thus, there is a gap in the research on the effect of phosphogypsum on pathogenic microorganisms, and helminth eggs and larvae, during the biological treatment of organic waste. Compared to previous studies that focus primarily on the effect of phosphogypsum on SRB growth, our study presents a novel aspect by investigating the indirect inhibitory effect of phosphogypsum on nematode egg viability, specifically through the formation of hydrogen sulfide during anaerobic digestion.

Therefore, this article focuses on studying the impact of a sulfur-containing additive, specifically phosphogypsum, on the viability of *Trichuris* nematode eggs in the treatment of pig manure.

## 2. Materials and Methods

**The characteristics of the substrate and inoculum for the biogas reactor.** The substrate and inoculum for the experiment were selected on the territory of a pig farm in Dnipro Region (LLC “Agroind”, Pidgorodne, Ukraine). Pig feces were used as a substrate, and fermented mass from an operating methane tank was used as an inoculum. Excrement and inoculum sampling was carried out under strict aseptic conditions to avoid contamination during all laboratory procedures [37]. The substrate and inoculum were placed in a sealed glass container with a volume of 2 L and transported to the laboratory of parasitological research of the Parasitology, Veterinary and Sanitary Expertise Department of the Dnipro State Agrarian and Economic University.

To make the digestate, 120 mL of inoculum, 120 mL of water, and 15 g of pig feces were poured into each biogas installation. As a sulfur-containing additive, phosphogypsum was added to the biogas reactors on the first day of the experiment at different concentrations (5%, 10%, 15%, and 20%) in relation to the substrate, and a control was also prepared (the inoculum and substrate without the addition of phosphogypsum) (Table 1).

The chemical composition of pig manure was as follows: N—6.2%; NH_4_N—1.7%; P_2_O_5_—5.55%; K_2_O—6.1%.

**The laboratory procedures.** A series of experiments on the effect of phosphogypsum on the manure disinfection process were carried out in two sequences, as shown in Figure 1.

**Testing the development of *Trichuris suis* eggs.** The study was conducted during August–November 2024 in two stages. In the first stage of the experiment, the effect of sulfur-containing additives, in particular, phosphogypsum, on the development of eggs was studied under aerobic conditions close to the conditions of natural development. A total of 0.1 mL of aqueous suspension of immature nematode eggs (without larvae) was placed in Eppendorf tubes (on average, 26 eggs), which were then filled with phosphogypsum solutions and left for 24 h at a temperature of 28 °C. After a day, the eggs were washed with water and cultivated for three months in a Micromed TC-20 thermostat (Micromed, Kharkov, Ukraine, 2023) at 28 °C. Four phosphogypsum solutions were prepared, 0.3%, 0.6%, 0.9%, and 1.2%, which corresponded to the amount of phosphogypsum added to the bioreactors in the second stage of the experiment (0.75, 1.50, 2.25 and 3.00 g to a total volume of 255 mL of digestate).

The second stage of the experiment was carried out under anaerobic conditions using the biogas reactors. This part of the experiment was carried out under mesophilic conditions (36 °C) on *T. suis* eggs. A total of 3 mL of an aqueous suspension with immature helminth eggs (without larvae) was placed in a biogas reactor for 25 days in an average amount of 540 ± 45 eggs/bioreactor.

The viability of the *T. suis* eggs [38] was measured by periodically selecting and examining 5 mL of digestate samples from each bioreactor 4 times: on the 5th, 10th, 20th, and 25th days of the experiment (until the biogas production process was completed). The eggs samples were then washed with water and filtered through sieves and settled using a centrifuge (Micromed, Kharkov, Ukraine, 2022), and then, the eggs were incubated for three months at a temperature of 28 °C using a Micromed TC-20 dry-air thermostat (Micromed, Kharkov, Ukraine, 2023).

Microscopy of the samples was performed using an Olympus DSX1000 microscope (Olympus Corporation, Tokyo, Japan, 2022). The degree of egg development was determined by changes in the internal structure of the eggs: the absence of changes, the presence of a cleavage process, or the presence of a formed larva. Morphometric characteristics were studied using the microscope camera (Micromed, Kharkov, Ukraine, 2021).

The data were processed using standard methods of variational statistics: the median, first and third quartiles, and minimum and maximum values were calculated. Samples were compared using one-way analysis of variance (ANOVA) and Tukey’s test (in the figures, samples that are significantly different from each other are marked with different letters). Differences were considered significant at *p* < 0.05.

**The laboratory procedure for anaerobic digestion.** Each bioreactor was a sealed polystyrene container with a sealed tap in the lower part to remove the liquid phase during digestion (Figure 2: 2). In the upper part of the bioreactor, there was a pipe (Figure 2: 3 and 4) for the gas phase (biogas) to be discharged into a gas collection bag (Figure 2: 5).

A five-step experiment was conducted with an exposure of 25 days. The experiment was carried out at a temperature of 36 °C. To maintain a constant temperature, a Micromed TC-20 dry-air thermostat (Micromed, Kharkov, Ukraine, 2023) was used. Once a day, the contents of the biogas reactors were mechanically stirred. The typical chemical composition [38] of manure in Ukraine corresponded to the chemical composition of the test samples used in our experiment (content at natural humidity of manure): total nitrogen—0.82–0.86%; ammoniacal nitrogen—0.14–0.16%; phosphorus—0.56–0.59%; C:N—(12.7 ± 0.4):1.

**Measurement of pH, CH_4_, and H_2_S.** On the first and last days of the experiment, a control measurement of pH was performed using an AD 1030 ADWA pH meter (Micromed, Kharkov, Ukraine, 2022).

Measurement of the main gas (CH_4_) formed in the installations was not performed on the first day of the experiment, since gas was not yet observed in the packages at this stage. On the 20th day of the experiment, a measurement of CH_4_ and H_2_S was performed using a Multirae gas analyzer (Honeywell RAE Systems, Sunnyvale, CA, USA, 2023).

## 3. Results and Discussion

During the first stage of the experiment to determine the effect of sulfur-containing additives, in particular, phosphogypsum, on the development of nematode eggs under conditions close to natural ones, it was found that the addition of phosphogypsum (Figure 3) to the aquatic environment with *Trichuris* eggs did not affect their survival (*p* = 0.643; F = 0.64, F_0.05_ = 2.87).

Figure 4 shows the stages of development of *Trichuris* eggs, which are characterized by the presence of a thick shell with a smooth surface, transparent polar plugs, a lemon shape, and brown color. It shows images of an immature egg in the protoplast (a) and cleavage stages (b), and a mature egg with a formed larva (c).

In the second stage of the experiment, when determining the effect of changes in internal conditions during anaerobic digestion with the addition of different concentrations of phosphogypsum on the development of nematode eggs, it was found that the pH value was constantly within the alkaline range—7.29–7.76 (Figure 5). In the control (without the addition of phosphogypsum), the pH value was slightly higher, which has also been confirmed by other authors’ studies on the initial pH values and the actual regulation of the acid–base balance during anaerobic digestion [39,40,41]. During our investigation, it was determined that at the concentrations of phosphogypsum used, the pH fluctuated slightly, decreasing toward an acid reaction, but remained, on average, below 7.

At the same time, the effect of adding phosphogypsum at certain concentrations on changes in CH_4_ in bioreactors under anaerobic conditions was studied (Figure 6), which proved to be consistent with previous studies [42].

With an increase in the experimental concentrations of phosphogypsum, the amount of CH_4_ decreased. In the control (without the addition of phosphogypsum), the amount of CH_4_ was, on average, 47.7%. With an increase in the amount of sulfur-containing additive, this indicator significantly decreased, on average, from 31.8% to 4.2%. This is in conformity with the studies of other authors. Thus, according to Khan and Trottier [43], the increase in the concentration of sulfur-containing compounds during anaerobic digestion is accompanied by the incorporation of the CH_4_ formation process. The inhibition of the process of methane formation in the presence of sulfur-containing substances was also described in experiments by Maslova et al. [44]. Mutegoa and Sahini [45] also reported that several other undesirable intermediate products are formed during anaerobic digestion, including hydrogen sulfide, which reduces the quality and quantity of the collected biogas. In order to enhance the production of short-chain fatty acids from the anaerobic digestion of activated sludge waste, Cheng et al. [46] proposed a new strategy based on the pretreatment of animal excrement with thiosulfate. However, these researchers noted that methanogenesis was significantly inhibited by such treatment. Accordingly, this should be taken into account when determining the appropriate dosage of phosphogypsum to avoid a significant reduction in methane content in the biogas. In future research, this aspect will be explored separately to minimize the risks associated with the toxic effects of sulfur compounds (H_2_S) on the methanogenesis process.

On the 5th day of the experiment, the mortality of eggs in the control did not exceed 30% compared to the 10th and 20th days, where this indicator significantly increased and was already, on average, over 80% (Figure 7). In the fourth week of the experiment (day 25), viable *Trichuris* eggs were detected neither in the control (without the addition of phosphogypsum) nor in other biogas reactors where the environmental conditions were changed by adding a sulfur-containing substance. In bioreactors where a sulfur-containing substance (phosphogypsum) was previously added, the mortality of eggs compared to the control increased significantly (up to 80% or more), even at the minimum concentration of phosphogypsum (5%), starting from the fifth day of the experiment. However, the maximum effect (100% mortality of *Trichuris* eggs) was observed at a 10% concentration of phosphogypsum starting from the 10th day of the experiment.

The addition of phosphogypsum during the anaerobic digestion of pig manure significantly accelerated the death of nematode eggs, in particular, of the genus *Trichuris*, from 25 to 10 and 5 days: 100% and 80% of the eggs were dead, on average, respectively. When obtaining valuable organic fertilizers, the optimal concentration of phosphogypsum is 10%, since on the 10th day, 100% death of nematode eggs was already recorded compared to the control (without the addition of phosphogypsum), where the mortality of eggs on the 10th day of the experiment did not exceed 85%.

In our experiment on the effect of phosphogypsum on the growth of nematode eggs in vitro, no developmental delay or death was recorded in most nematode eggs. However, previous studies [42] have already investigated the effect of different doses of phosphogypsum on the composition of biogas in the anaerobic digestion process, and it was substantiated that an increase in hydrogen sulfide concentration was observed. This, in turn, is due to the fact that phosphogypsum components are used by sulfate-reducing bacteria that produce hydrogen sulfide, which was investigated in [47].

Therefore, based on previous studies [42,48], the current study substantiates that phosphogypsum indirectly inhibits the viability of nematode eggs by generating extra hydrogen sulfide during the addition of phosphogypsum under anaerobic digestion. In the current series of experiments with phosphogypsum addition, H_2_S concentrations as high as 5000 ppm were observed. Therefore, sulfate reducers use sulfates as an electron acceptor in their own metabolism, and accordingly, the introduction of phosphogypsum stimulates their development [49,50]. The effect of hydrogen sulfide on nematode eggs has been described in works by other researchers [48,51]. Hurley and Sommerville [48], in their experiments on embryonated nematode eggs (*Ascuris suum*), found that hydrogen sulfide affects the egg shell, combining with some of its components and stimulating the hatching of infective eggs (more than 85%).

Summarizing the results of our own research and the research of other authors [21,22], a comprehensive scheme of the influence of phosphogypsum on the disinfection of livestock waste (manure, etc.) in the process of anaerobic digestion under sulfate reduction conditions was developed, which is shown in Figure 8.

Therefore, it is necessary to further integrate the complex treatment of waste from livestock complexes into two-stage bioreactors, which will allow optimization of the process of waste disinfection in sulfate reduction conditions under the influence of phosphogypsum additive, as well as mitigate the impact on methanogenic archaea in the future.

## 4. Conclusions

As a result of this investigation, the possibility of using phosphogypsum for the process of the decontamination of organic waste was identified under the stimulation of sulfate reduction. This made it possible to obtain safe and valuable organic fertilizers (without viable *Trichuris suis* eggs) and, at the same time, use a more energy-efficient mesophilic mode of the bioreactor operation. Changes in anaerobic digestion conditions using a sulfur-containing additive could positively affect digestate decontamination, accelerating this process. However, an increase in the content of hydrogen sulfide under the conditions of stimulating sulfate reduction is necessary, which affects the qualitative and quantitative composition of biogas. It is necessary to take into account the target orientation of organic waste processing. If the primary task is to obtain high-quality environmentally safe fertilizers, then anaerobic digestion under sulfate reduction conditions is a promising direction for processing. At the same time, further research is needed to optimize the conditions of the anaerobic process for effective disinfection during the formation of organic fertilizers and the simultaneous production of valuable biofuel. The possibility of using phosphogypsum as an additive to digestate during its storage and subsequent application is also planned for a future investigation.

## Figures and Tables

**Figure 1 microorganisms-13-01165-f001:**
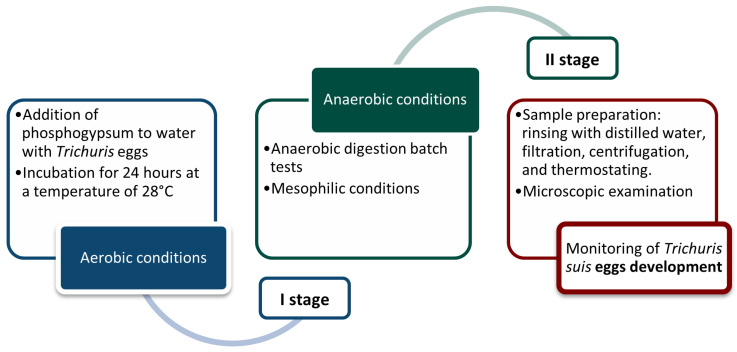
The stages of the experiments to study the effect of phosphogypsum on disinfection in the process of anaerobic digestion.

**Figure 2 microorganisms-13-01165-f002:**
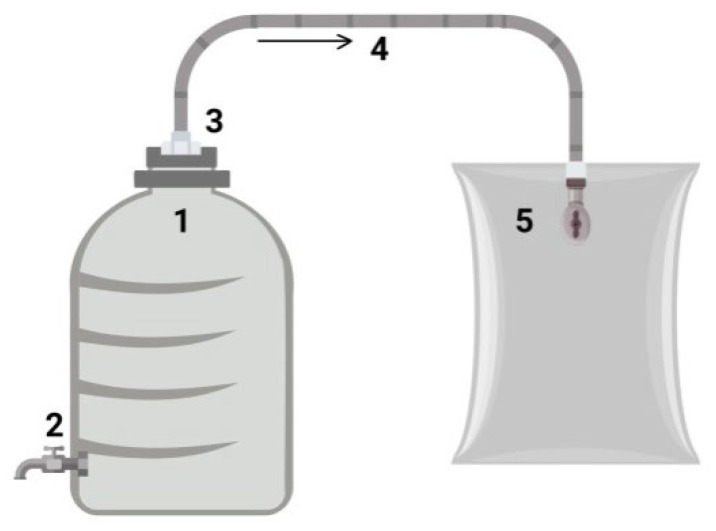
Anaerobic digestion lab bench: 1—hermetic polystyrol; 2—hole with sealed valve part; 3—connection of hermetic polystyrol with outlet pipe; 4—outlet pipe; 5—gas-harvesting reservoir.

**Figure 3 microorganisms-13-01165-f003:**
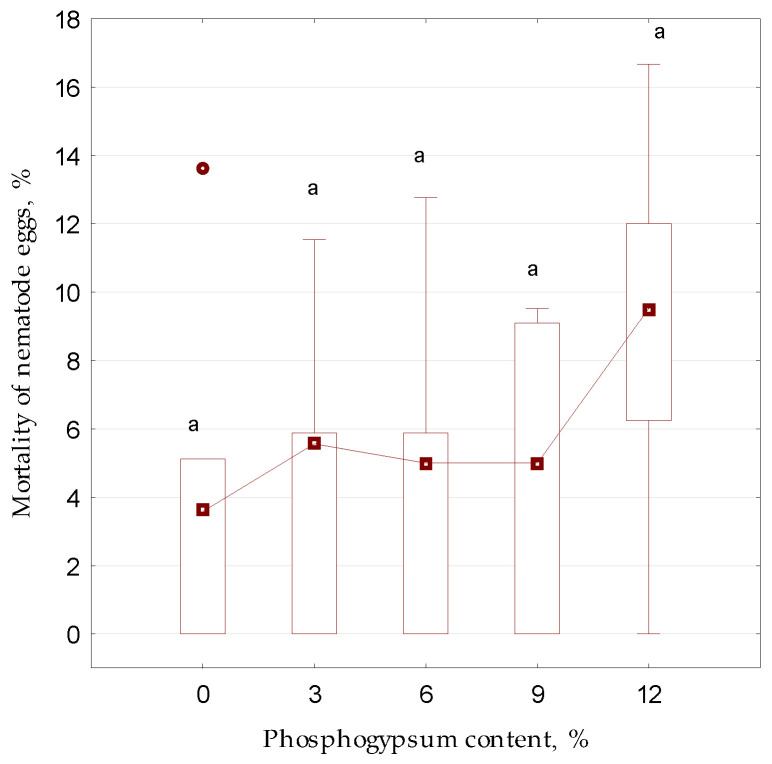
The effect of sulfur-containing additive (phosphogypsum) on the mortality of *Trichuris suis* eggs during a 90-day in vitro laboratory experiment under aerobic conditions (*n* = 5): different letters above the samples in this figure and below correspond to reliable differences (*p* < 0.05) between them according to the results of Tukey’s test; since no reliable differences were registered between the samples, the letter “a” is indicated above each sample; small square—median; small circle—outlier; lower and upper limits of the rectangle—first and third quartiles; respectively; lower and upper limits of the vertical line—minimum and maximum values.

**Figure 4 microorganisms-13-01165-f004:**
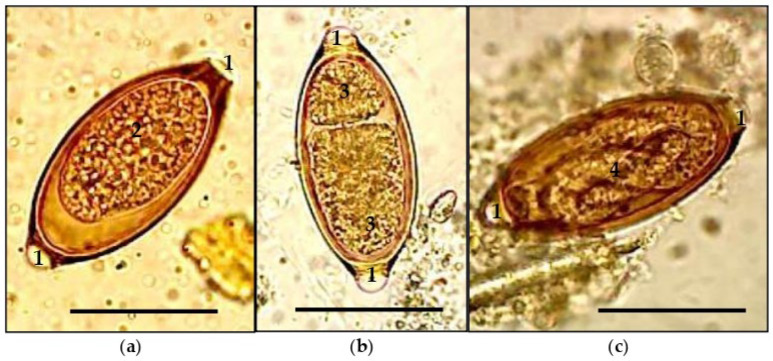
Stages of embryonic development of *Trichuris* eggs: (**a**)—protoplast; (**b**)—cleavage stage; (**c**)—larval stage. Length of black bar—40 μm. 1—cork; 2—protoplast; 3—blastomere; 4—larva.

**Figure 5 microorganisms-13-01165-f005:**
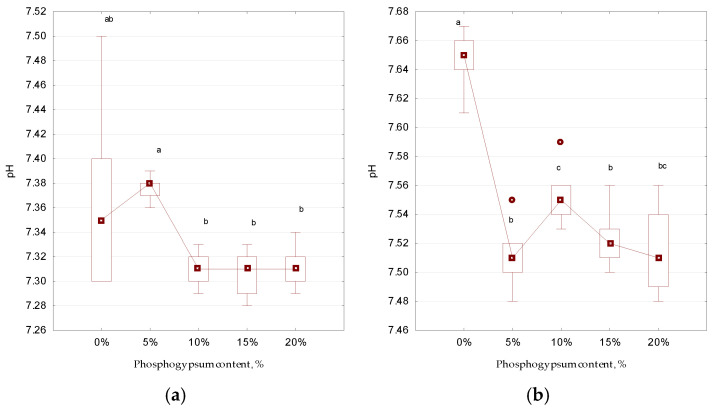
The effect of sulfur-containing additives on the change in pH during anaerobic digestion (*n* = 5): (**a**)—day 1, (**b**)—day 25; different letters above the samples in this figure and below correspond to reliable differences (*p* < 0.05) between them according to the results of Tukey’s test; for designation, see Figure 3.

**Figure 6 microorganisms-13-01165-f006:**
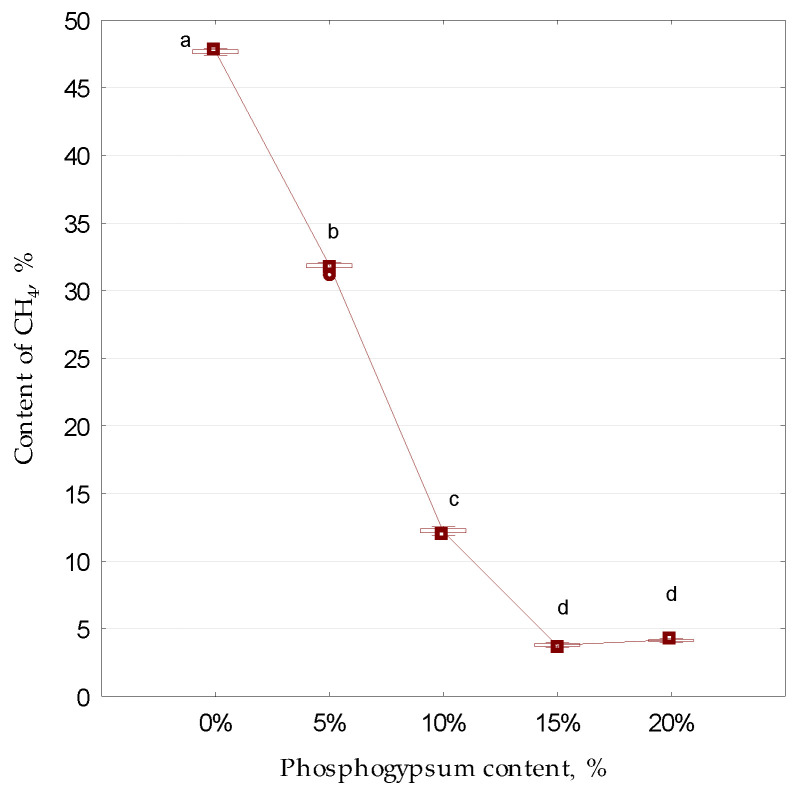
The effect of sulfur-containing additives on the change in the CH_4_ content in the gaseous medium of bioreactors during anaerobic digestion (*n* = 5): different letters above the samples in this figure and below correspond to reliable differences (*p* < 0.05) between them according to the results of Tukey’s test; along the abscissa axis—the content of phosphogypsum in the medium (%);along the ordinate axis—the content of CH_4_ (%); for designation, see Figure 3.

**Figure 7 microorganisms-13-01165-f007:**
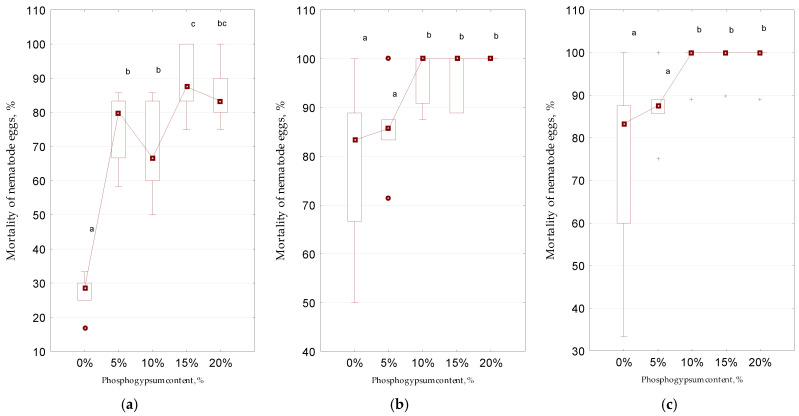
Effect of sulfur-containing additive (phosphogypsum) on mortality of *Trichuris suis* eggs during anaerobic digestion in biogas installations (*n* = 5): (**a**)—day 5; (**b**)—day 10; (**c**)—day 20; +—extreme; different letters above the samples in this figure and below correspond to reliable differences (*p* < 0.05) between them according to the results of Tukey’s test; for designation, see Figure 3.

**Figure 8 microorganisms-13-01165-f008:**
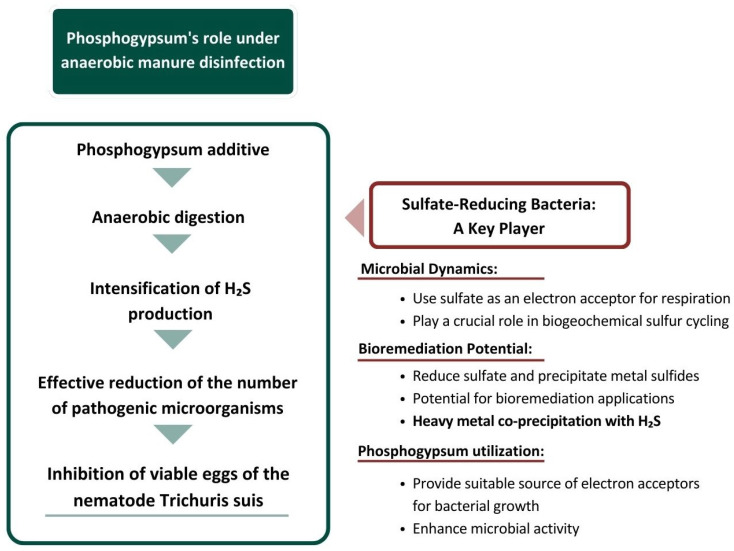
Anaerobic digestion under sulfate reduction conditions: phosphogypsum as an influencing factor.

**Table 1 microorganisms-13-01165-t001:** Concentrations of major elements in phosphogypsum.

Element	CaO	SO_2_	SiO_2_	Al_2_O_3_	P_2_O_5_	Fe_2_O_3_	K_2_O	TiO_2_	Na_2_O	MnO	MgO
Concentrations, wt.%	22.9–31.4	29.8–36.0	13.1–24.7	0.96–2.52	0.63–0.79	0.41–0.94	0.10–0.32	0.05–0.17	0.02–0.07	0.01	0.01

## Data Availability

The original contributions presented in this study are included in the article. Further inquiries can be directed to the corresponding authors.

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
