# Peer review of "Impact of Phosphogypsum on Viability of Trichuris suis Eggs in Anaerobic Digestion of Swine Manure"

_microorganisms, 2025, doi:10.3390/microorganisms13051165_

Round 1

Reviewer 1 Report

Comments and Suggestions for Authors

The author was focus on the effect of sulfur containing additives such as phosphonyls on swine manure Trichuris suis eggs. To make it easier for the reader to understand the author depicts a schematic diagram of the experiment, which is the strength of this article. But there are still some places in the article that are not clearly described.

Title: 
This study mainly focusses on the effect of sulfur containing additives such as phosphonyls on swine manure Trichuris suis eggs. In the title should be more precise about the subject of the study, the current title is too generalized.

Line 209: 
“Trichuris” should be italic format.
Figure 3: 
what is the little circle mean? Author should describe it in the figure legend. 
Figure 5: 
Y-axis: There should be a decimal point instead of a comma between two numbers.
Figure 7c: 
What is “※” means? Author should describe it in the figure legend. 

Author Response

Dear Reviewer!

Thank you for your help in improving the quality of our manuscript! We have tried to take into account all your comments and wishes.

Comments 1: This study mainly focusses on the effect of sulfur containing additives such as phosphonyls on swine manure Trichuris suis eggs. In the title should be more precise about the subject of the study, the current title is too generalized.

Response 1:

Thank you for your comment, we have made adjustments and clarifications to the title of the article: ‘Impact of Phosphogypsum on the Viability of Trichuris suis Eggs in Anaerobic Digestion of Swine Manure’

Comments 2: Line 209: “Trichuris” should be italic format.

Response 2:

Thank you. Corrected.

Comments 3: Figure 3: what is the little circle mean? Author should describe it in the figure legend. 

Response3:

Thank you. Corrected.

Comments 4: Figure 5: Y-axis: There should be a decimal point instead of a comma between two numbers.

Response 4:

Thank you. Corrected.

Comments 5: Figure 7c: What is “※” means? Author should describe it in the figure legend. 

Response 5:

Thank you. Corrected.

Dear Reviewer 1,

Thank you again for your time and dedication to review our manuscript and make it better.

Reviewer 2 Report

Comments and Suggestions for Authors

In the current MS, the Authors investigated the effect of phosphogypsum on the viability of nematode’s eggs. The authors showed that adding phosphogypsum to pig manure reduces the viability of Trichuris nematode eggs. The main part of the work is devoted to parasitology and, in my opinion, does not correspond the goals and aim of the journal. Although the authors discuss increasing methane content during anaerobic digestion, the authors did not conduct any experiments with microorganisms.

Minor

Line 177-181. What software was used to calculate the statistics? Please, add this info in MS.

Line 191. Authors used 36C. But in Introduction section authors mentioned about different temperatures but not about 36. Please, could Authors explain the chosen temperature?

Line 219: “protoplast” – is standard term for immature eggs of nematods?

Figure 4. Please, marks all visible structures on A-C images. In head of Fig authors should add “light images”

Line 223: replace “black” to “scale”

Line 235: “designation see Fig. 3.” Although the authors used similar designations for Fig.5 as for Fig.3, the designations for Figure 5 should be set again. The same for line 246-247 and 265-266

Line 260-262. However, this sentence is for a Discussion section. After all, in the current work, the authors did not study what happens to the parasite shell. I recommend moving this proposal to the Discussion section.

Line 261: “shell” – is available term for envelop of parasite eggs? Usually shell is used for mollusk.

Line 270: The authors decelerate that the death of nematode eggs was at three points (25, 10 and 5 days), but only two values were set «100 and 80%. The value for 10 or 5 is missing. Please set the missing value.

Line 277: “does not directly kill the parasites” What authors mean? А как убивал phosphogypsum? Please, explain

Line 279 “sulfate-reducing bacteria” – in the current study, the authors did not show the presence of sulfate-reducing bacteria. Therefore, it is not necessary to say directly that sulfate-reducing bacteria that led to the appearance of hydrogen sulfide. Please rephrase this sentence.

Line 284-300. In this paragraph, the authors mentioned literature data on the increase in the concentration of sulfur-containing compounds during anaerobic fermentation. The authors should not only cite literature data, but also discuss the data obtained in current research. The authors should either discuss the data obtained or delete this paragraph, or move it in Introduction section.

Line 301-317. The authors provide a literature review of methods for increasing methane content during anaerobic digestion. However, the authors do not discuss the data which they obtained in the current study. In current form, this paragraph should be removed from the MS. Or Authors should discuss own data.

Main

In Materials and Methods section Authors have to add Country and manufacture for each used equipment: biogas reactor, centrifuge, thermostat etc. Authors should remember that the country and manufacturer of the equipment should be mentioned only once, the first time mentioned.

Line 282-283. The authors contradict themselves. At the beginning of the paragraph, the authors state that “no developmental delay was recorded.” And in line 282-283, the authors conclude that “negatively affects the development of nematode eggs.” The authors should revise this paragraph and rewrite it according to the data obtained in the current study.

Author Response

Dear Reviewer,

Thank you very much for taking the time to review this manuscript. We greatly appreciate all of your constructive comments and suggestions. We have done our best to address all your suggestions and have provided detailed responses in a separate file for your convenience.

Reviewer 3 Report

Comments and Suggestions for Authors The authors set the stage to investigate the effect of sulfur-containing additive such as phosphogypsum on the disinfection of pig manure during anaerobic fermentation. The results showed that egg mortality increases significantly compared to the control sample (80% and more), even at a minimum concentration of phosphogypsum (5 %) already on the 5th day of the experiment. The maximum effect  (100 % mortality) was observed at a 10 % concentration of phosphogypsum. This study revealed that changes in anaerobic digestion conditions using phosphogypsum could positively affect digestate disinfection. It identifies the possibilities of using phosphogypsum for the process of decontamination of organic waste under conditions of stimulation of sulfate reduction. There are some issues that have to be addressed to this research work in order to be improved.   1) In introduction, the authors should define the novelty of their study compared to literature-based studies.   2) In the laboratory procedure for anaerobic fermentation, the authors should describe the five experiments were conducted (e.g composition, temperature)   3) How the sulfur-containing additives affect the content of methane produced at the 1st day, the 5th day and at the 25th day (a diagram or a table with methane production values at different days and at different phosphogypsum content)?         4) Have the authors investigated the microbial community for different contents of phosphogypsum?

Author Response

Thank you very much for taking the time to review this manuscript. We greatly appreciate all of your constructive comments and suggestions. We have done our best to address all your suggestions and have provided detailed responses in a separate file for your convenience.

Round 2

Reviewer 2 Report

Comments and Suggestions for Authors

Authors implementd all of my concernss/propositions, and now MS is look good. But my main concerns is still unanswered - "The main part of the work is devoted to parasitology and, in my opinion, does not fit the goals and objectives of the journal". The MS looks good , but I still have to reject it, since that the work does not correspond to the goals and aim of the Microorganisms journal.  I recommend to submit MS in more suitable journal. 

P.S. May be Authors should contact with editor to find out whether their work correspond to aim and goal of Microorganisms journal. 

Reviewer 3 Report

Comments and Suggestions for Authors

-